# SpikeGPT: Generative Pre-trained Language Model with Spiking Neural Networks

**Rui-Jie Zhu**                                                              *rzhu48@ucsc.edu*
*Department of Electrical and Computer Engineering*
*University of California, Santa Cruz*

**Qihang Zhao**                                                    *zhaoqihang@kuaishou.com*
*Kuaishou*

**Guoqi Li**                                                                  *guoqi.li@ia.ac.cn*
*Institute of Automation*
*Chinese Academy of Sciences*

**Jason K. Eshraghian**                                                         *jsn@ucsc.edu*
*Department of Electrical and Computer Engineering*
*University of California, Santa Cruz*

**Reviewed on OpenReview:** *https://openreview.net/forum?id=gcf1anBL9e*

## Abstract

As the size of large language models continue to scale, so does the computational resources required to run them. Spiking Neural Networks (SNNs) have emerged as an energy-efficient approach to deep learning that leverage sparse and event-driven activations to reduce the computational overhead associated with model inference. While they have become competitive with non-spiking models on many computer vision tasks, SNNs have proven to be more challenging to train. As a result, their performance lags behind modern deep learning, and until now, SNNs have yet to succeed at language generation on large-scale datasets. In this paper, inspired by the Receptance Weighted Key Value (RWKV) language model, we successfully implement 'SpikeGPT', a generative language model with binary, event-driven spiking activation units. We train the proposed model on two model variants: 46M and 216M parameters. To the best of our knowledge, SpikeGPT is the largest backpropagation-trained SNN model when released, rendering it suitable for both the generation and comprehension of natural language. We achieve this by modifying the transformer block to replace multi-head self-attention to reduce quadratic computational complexity $\mathcal{O}(T^2)$ to linear complexity $\mathcal{O}(T)$ with increasing sequence length. Input tokens are instead streamed in sequentially to our attention mechanism (as with typical SNNs). Our experiments show that SpikeGPT remains competitive with non-spiking models on tested benchmarks, while maintaining $32.2\times$ fewer operations when processed on neuromorphic hardware that can leverage sparse, event-driven activations. Our code implementation is available at `https://github.com/ridgerchu/SpikeGPT`.

## 1 Introduction

Artificial Neural Networks (ANNs) have recently achieved widespread, public-facing impact in Natural Language Processing (NLP), but with a significant computational and energy consumption burden across training and deployment. As an example, training GPT-3 was projected to use 190,000 kWh of energy (Brown et al., 2020; Dhar, 2020; Anthony et al., 2020). Spiking neural networks (SNNs), inspired by neuroscientific

models of neuronal firing, offer a more energy-efficient alternative by using discrete spikes to compute and transmit information (Maass, 1997). Spike-based computing combined with neuromorphic hardware holds great potential for low-energy AI (Davies et al., 2018; Merolla et al., 2014; Sun et al., 2022; Nath et al., 2024), and its effectiveness in integration with deep learning has been demonstrated through numerous studies (Roy et al., 2019; Pfeiffer & Pfeil, 2018; Fang et al., 2021; Eshraghian et al., 2021; Wu et al., 2018; Zhang et al., 2020).

While SNNs have shown competitive performance in computer vision tasks such as classification and object detection (Barchid et al., 2023; Kim et al., 2020; Cordone et al., 2022), they have yet to attain similar success in generative models. With respect to language generation, this can be attributed to several reasons: i) the absence of an effective language encoding technique for SNNs, ii) the difficulty of training large-scale SNNs due to the extreme constraint on layer-to-layer bandwidth (i.e., binarized spike activations) (Eshraghian et al., 2022), and the lack of informative gradient signals in excessively sparsified models (Eshraghian & Lu, 2022), and iii) the inherent recurrence of SNNs is incompatible with self-attention, where input token are passed to the model in parallel (Vaswani et al., 2017). These issues mean that training large-scale SNNs via error backpropagation is extremely challenging, leading to a lack of performant SNNs in language generation.

Despite the difficulties faced by recurrent networks in NLP, the sequential structure of linguistic data presents a unique advantage for SNNs. To address these problems, we propose three techniques. First, rather than using an encoder to project information into an additional temporal dimension, SpikeGPT aligns the sequence dimension of language with the temporal dimension of SNNs. This eliminates the need for an encoder as with most prior attention-based SNNs (Zhou et al., 2023; Li et al., 2022; Deng et al., 2024; Qiu et al., 2024). Second, we apply autoregressive training rather than accumulating spikes at the output to calculate the loss, as is common in modern SNNs. Third, although binary activations constrain layer-to-layer bandwidth, we show that using stateful neurons can greatly reduce the adverse impact of binarization. By leveraging the capabilities of the RWKV model and integrating these three techniques, we have developed the SpikeGPT language model. This was the first SNN to achieve language generation at the time of writing and code release of SpikeGPT. It is also the largest SNN trained to date in terms of parameter count when it is released; the largest version has 216M parameters, which is $3\times$ more than the previous largest SNN (Zhou et al., 2023).

The implementation of SpikeGPT is based on integrating recurrence into the Transformer block such that it is compatible with SNNs and eliminates quadratic computational complexity, allowing for the representation of words as event-driven spikes. Combining recurrent dynamics with linear attention enables our network to stream incoming data word-by-word, and commence computation before a sentence has been completed, while still retaining long-range dependencies present in complex syntactic structures. Our experiments show that SpikeGPT achieves competitive performance on all tested datasets while consuming significantly less energy compared to traditional ANN models. Our contributions in the field of NLP and language generation can be succinctly described as follows:

1. we provide the first demonstration of language-generation using direct-SNN training;

2. we achieve performance comparable to that of ANNs, while preserving the energy efficiency of spike-based computations and reducing quadratic computational complexity $\mathcal{O}(T^2)$ to linear complexity $\mathcal{O}(T)$;

3. our results demonstrate that a small-scale variant of the SpikeGPT model with 46 million parameters performs competitively against similar transformer models. Furthermore, it achieves this performance with an estimated $33.2\times$ less energy consumption on asynchronous hardware.

## 2 Related Works

Although language generation has not previously been achieved with SNNs, this section provides an overview of how SNNs have been used in basic NLP tasks, and the ways in which transformers have been adopted for SNNs.

**Spiking Neural Networks for Natural Language Processing.** Xiao et al. (2022) proposes a bidirectional SNN for sentiment classification and machine translation tasks. Their approach uses spiking

encoders, which replace costly multiplication operations with much cheaper additive operations to significantly reduce computational energy consumption. Similarly, Lv et al. (2023b) presents a two-step method to train SNNs for text classification with a simple way to encode pre-trained word embeddings as spike trains. Their results indicate that converted SNNs achieve comparable results to their ANN counterparts and are more robust against adversarial attacks. Furthermore, Diehl et al. (2016) demonstrate the train-and-constrain methodology that enables the mapping of machine-learned recurrent neural networks (RNNs) on a substrate of spiking neurons. The authors achieve 74% accuracy on a question classification task using less than 0.025% of the cores on one TrueNorth chip (Merolla et al., 2014), showcasing the potential for SNNs in classification tasks in NLP. For the pre-train language model side, two works ( Bal & Sengupta (2023) and Lv et al. (2023a)) offer a direct-trained spiking Bidirectional Encoder Representations from Transformers (BERT) model via knowledge distillation.

**Transformer in Spiking Neural Networks.** The Transformer model, first introduced in Vaswani et al. (2017), has shown significant success in various NLP tasks. However, the application of the Transformer model to SNNs has been relatively limited. The first Spiking Transformer model was proposed in Zhou et al. (2023), which proposes spiking self-attention to model visual features using sparse Query, Key and Value matrices. Li et al. (2022) proposes another variant on Transformer-based SNNs, adopting spatial-temporal attention instead of spatial or temporal-wise attention to better incorporate the attention mechanism within the Transformer.

While Transformers were initially proposed to solve NLP tasks, SNN-based Transformers have only succeeded with vision tasks. An additional temporal dimension must be added which increases the computational complexity from quadratic to the cubic order ($\mathcal{O}(T^3)$), which makes training more expensive. The additional challenges of extreme sparsity, non-differential operators, approximate gradients, and single-bit activations that are characteristic of SNNs make training convergence more challenging. The demonstrated image classification tasks have a far smaller number of output classes, which shrinks the scale of demonstrated networks. Image classification also does not exploit the inherent long-range learning capacity of self-attention. Therefore, there is under-explored potential in the application of Transformer models in other SNN-based applications beyond vision tasks. In the following sections, we demonstrate how this computational complexity is reduced to enable scaled-up models that are capable of language generation.

## 3 Methods

### 3.1 Leaky Integrate-and-Fire Neuron

We employ the Leaky Integrate-and-Fire (LIF) neuron as the default spiking neuron of our model, a widely used model for SNNs often trained via error backpropagation (Maass, 1997). The LIF dynamics are represented as follows:

$$\begin{cases} U[t] = H[t] + \beta(Y[t] - (H[t-1] - U_{\text{reset}})) \\ S[t] = \Theta(U[t] - U_{\text{threshold}}) \\ H[t] = U[t] \cdot (1 - S[t]) \end{cases} \tag{1}$$

where $\beta$ is a decay factor, $U$ is the membrane potential (or hidden state) of the neuron, $S$ is the spiking tensor with binarized elements, $Y$ denotes the output of the previous series RWKV block (see Eq. 7), $\Theta(\cdot)$ denotes the Heaviside function, and $H$ represents the reset process after spike emission. We set $U_{\text{threshold}} = 1$, $U_{\text{reset}} = 0$ and $\beta = 0.5$ as done in Zhu et al. (2022) and Yao et al. (2021; 2023).

To overcome the non-differentiable problem during back-propagation caused by the Heaviside step function $\Theta(\cdot)$, we employ the arctangent surrogate function during the backward pass. The arctangent function $\sigma'(x) = \frac{\alpha}{2(1+(\frac{\pi}{2}\alpha x)^2)}$ (a Sigmoid-like shape) is applied as a 'surrogate gradient' for backward propagation to provide a biased gradient estimator (Fang et al., 2021; Neftci et al., 2019). Additional details on the derivation and training of LIF neurons using backpropagation are provided in Appendix A.

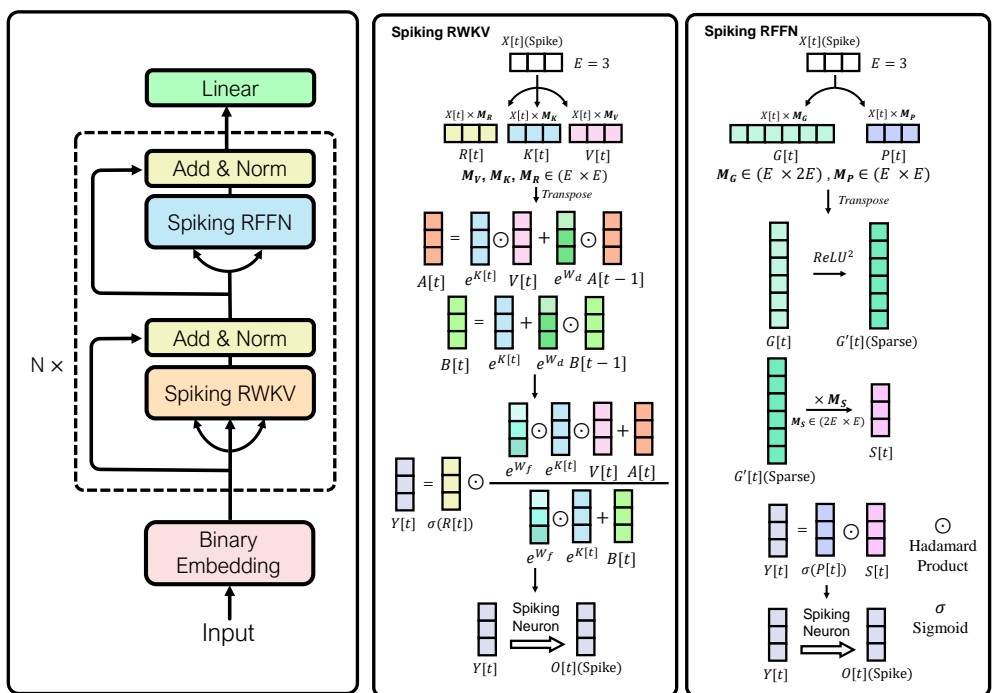

Figure 1: Model Architecture. The left portion displays the block-level structure. The middle and right illustrations demonstrate the Spiking RWKV and Spiking RFFN architectures, respectively. Spiking RWKV serves as a token mixer and Spiking RFFN functions as a channel mixer. These components are arranged in a loop with residual connections in a manner akin to a Transformer architecture.

## 3.2 Model Architecture

The high-level architecture of SpikeGPT is shown in Fig. 1. Given an embedded input $I \in \mathbb{R}^{T \times d}$, we first use a Binary Embedding (BE) layer to embed the input to a binarized representation, $X_0$ (Eq. 2). Using Spiking RWKV, $X_0$ is passed to the L-*th* SpikeGPT layer. Similar to the standard Transformer block, a SpikeGPT block consists of Spiking RWKV (SRWKV) unit and a Spiking Receptance Feed-Forward Networks (SRFFN) unit. Residual connections are used in both the SRWKV and SRFFN blocks using the SEW-ResNet (Fang et al., 2021) configuration, which is a well-established standard form within the Spiking ResNet framework with sparse integer spikes. Once the data has traversed through all layers, the model is directed towards the generation head (GH) for the purpose of generating the next token. For natural language understanding (NLU) tasks, the model uses a classification head instead. Sec. 3.6 provides further information.

$$X_0 = \text{BE}(I), \qquad\qquad I \in \mathbb{R}^{T \times d}, X_0 \in \mathbb{R}^{T \times d}, \tag{2}$$

$$X'_l = \text{SRWKV}(X_{l-1}) + X_{l-1}, \qquad\qquad X'_l \in \mathbb{R}^{T \times d}, l = 1...L \tag{3}$$

$$X_l = \text{SRFFN}(X'_l) + X'_l, \qquad\qquad X_l \in \mathbb{R}^{T \times d}, l = 1...L \tag{4}$$

$$Y = GH(X_L). \tag{5}$$

In Section 3.3, detailed information about Binary Embedding (BE) will be provided. Sec. 3.4 will primarily focus on the SRWKV block, while Sec. 3.5 will delve into the Spiking Receptance Feed-Forward Network (SRFFN) block. Finally, in Sec .3.6, we will describe the training and inference details employed for natural language generation (NLG) and natural language understanding (NLU). At the commencement of each block, a token-shift is applied which is detailed in Appendix C.1.

### 3.3 Binary Embedding

To maintain consistency with the binary activations of SNNs, we propose a binary embedding step to convert the continuous outputs of the embedding layer into binary spikes. The conversion is performed using a Heaviside function for feed-forward propagation which maps the continuous values to binary spikes. This allows us to convert continuous embedding values into spikes using non-differentiable functions, while still being able to perform backpropagation and update the weights of the embedding layer (Neftci et al., 2019). During backpropagation, the arctangent surrogate function is applied as is the case with LIF neurons.

### 3.4 Efficient Processing of Variable-Length Sequences Using Spiking RWKV

In this section, we revisit the self-attention mechanism, which endows Transformers with the ability to process variable-length sequences, enabling advanced language modeling capabilities. Subsequently, we highlight a fundamental challenge: the incompatibility of self-attention with SNN language modeling. Self-attention relies on access to the entire sequence for effective language modeling and is not inherently recurrent, making it incompatible with SNNs. Furthermore, it involves dynamic matrix-matrix multiplication operations. The associated computational overhead compromises the objective of reducing computational costs via SNNs. To address these limitations, we introduce the Spiking RWKV (SRWKV) module as a replacement for the self-attention module. This module draws inspiration from the RWKV model (Peng et al., 2023), an RNN model renowned for achieving Transformer-level proficiency across a spectrum of model sizes. This module serves the same essential function of facilitating information exchange across token dimensions but achieves this through element-wise products rather than matrix-matrix multiplication, while also accommodating the recurrent nature of SNNs.

**Recall Self-Attention.** In the context of Transformers, the self-attention mechanism operates on an input sequence denoted as $X \in \mathbb{R}^{T \times d}$ and employs a scaled dot product attention technique. Mathematically, self-attention is formally expressed as follows:

$$f(X) = \text{softmax}\left(\frac{Q(K)^T}{\sqrt{d_k}}\right) V, Q = XM_Q, K = XM_K, V = XM_V \tag{6}$$

Here, $M_Q \in \mathbb{R}^{d \times d_k}$, $M_K \in \mathbb{R}^{d \times d_k}$, and $M_V \in \mathbb{R}^{d \times d_v}$ represent linear transformations, while $d_k$ and $d_v$ signify the dimensions of the key and value vectors, respectively. This mechanism enables dynamic information mixing across token dimensions and accommodates sequences of variable lengths by leveraging dynamic matrix-matrix multiplication $(Q(K)^T)$ and the softmax function (which necessitates access to the entire sequence).

Nevertheless, in the case of a typical SNN characterized by its event-driven nature, the recurrent structure of the network poses a challenge. SNNs can only generate spikes based on information from previous time-steps, making it impossible to access the entire sequence at once, as required by self-attention. One potential solution involves introducing an additional temporal dimension, denoted as $T_{\text{additional}}$, to allow spiking neurons to feed forward. However, this approach would substantially increase the model's size and necessitate additional computations, scaling proportionally with $T_{\text{additional}}$. This is not practical, especially for large-scale language models reaching the scale of hundreds of billions of parameters. Therefore, the optimal approach is not to introduce a new dimension, but to leverage the existing sequence dimension for forward propagation. This is why a spiking transformer would benefit from a recurrent alternative to self-attention.

**Spiking RWKV.** To address the challenges posed by self-attention in the context of language modeling, we introduce the Spiking RWKV, a novel approach that incorporates both recurrent and block-level spiking features. The foundation of the vanilla RWKV is inspired by the Attention Free Transformer (Zhai et al., 2021), serving as a replacement for the traditional self-attention mechanism. For comprehensive information on vanilla RWKV, including its overall structure and parallelization techniques, we refer the reader to Appendix C.1. In contrast to self-attention, which operates on the entire sequence dimension, the spiking input of Spiking RWKV is unrolled as $X[t] \in \mathbb{R}^{1 \times d}$, where $t$ represents the time step index, rather than the entire sequence $X \in \mathbb{R}^{T \times d}$. Similar to self-attention, Spiking RWKV initiates by applying linear transformations: $R = X[t]M_R$, $K = X[t]M_K$, and $V = X[t]M_V$, where $M_R, M_K, M_V \in \mathbb{R}^{d \times H}$, with $H$

indicating the hidden size. Notably, all inputs to the three MLP layers are in the form of spiking activations. The subsequent step involves generating the output using the following equation:

$$Y[t+1] = \mathcal{SN}\left(\sigma(R[t]) \odot \frac{\exp(W_f) \odot \exp(K[t]) \odot (V[t]) + A[t]}{\exp(W_f) \odot \exp(K[t]) + B[t]}\right) \tag{7}$$

The hidden states $A$ and $B$ are defined as:

$$A[t] = \exp(K[t]) \odot (V[t] + \exp(W_d) \odot A[t-1] \tag{8}$$

and

$$B[t] = \exp(K[t-1]) + \exp(W_d) \odot B[t-1] \tag{9}$$

Here, $\mathcal{SN}$ refers to spiking neuron layers, the $\odot$ symbol represents element-wise multiplication, $W_d \in \mathbb{R}^{1\times d}$ and $W_f \in \mathbb{R}^{1\times d}$ are decay vectors, where $W_f$ is responsible for weighting the current time-step's information, and $W_d$ plays a role in decaying the influence of information from previous time-steps. Details of $W_d$ and $W_f$ can be found in Appendix C.4. Notably, Spiking RWKV diverges from self-attention by not employing matrix-matrix multiplication to dynamically adjust the attention map according to the input. Instead, it utilizes the learnable vectors $W_d$ and $W_f$ to recurrently blend the token dimensions of the input. While self-attention dynamically reweights token dimensions based on the input, Spiking RWKV adopts a continuous decay strategy rather than a dynamic weighted attention strategy. Because the hidden states $A[t]$ and $B[t]$ encapsulate information from the previous time-step, this module can effectively blend information across token dimensions, much like self-attention. Additionally, this approach is particularly well-suited for language modeling. When using self-attention in casual language models, it's necessary to mask out half of the attention map to prevent information leakage. In contrast, Spiking RWKV combines the inherent recurrent properties of SNN with RWKV, and both naturally operate in a unidirectional manner.

### 3.5 Spiking Receptance Feed-Forward Networks (SRFFN)

Each block in our model contains a fully connected feed-forward network with a gating mechanism (SRFFN), which is applied to normalized and token-shifted outputs of each spiking-RWKV module. This SRFFN module consists of three linear transformations as follows:

$$Y'[t] = \mathcal{SN}(\sigma(M_P X[t]) \odot M_S(\text{Activation}(M_G X[t]))) \tag{10}$$

where $\mathcal{SN}$ represents spiking neuron layers, $Y'[t]$ denotes the output of SRFFN at time-step $t$ which is then passed to the spiking neuron (Eq. 1). $\{M_P, M_G, M_S\} \in \mathbb{R}^{d\times H}$ are learnable parameters of the linear transformations, and Activation represents the activation function. Depending on the specific SpikeGPT variant being used, we employ either the $ReLU^2$ or $\mathcal{SN}$ activation function. SRFFN is a variant of the Gated Linear Unit (GLU) (Dauphin et al.,

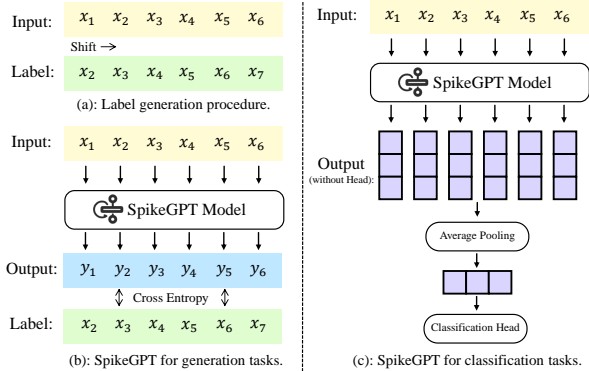

Figure 2: Training SpikeGPT for NLG and NLU tasks.

2017), which can control the degree of information flowing into the model by $\sigma(M_P X[t])$. In order to maintain consistency between SRFFN and GEGLU parameters (Shazeer, 2020), we set the size of $H$ from the SRFFN to $4d$. After the feedforward process in the SRFFN layer, the resulting output $Y'[t]$ will be subsequently fed into a LIF neuron, as depicted in Eq. 1. In this context, $Y[t]$ is replaced with the updated value $Y'[t]$ to preserve the block-level spiking and binary characteristics, thereby ensuring the maintenance of the desired sparsity.

### 3.6 Training & Inference

Our training procedure consists of two stages. The first stage is pre-training on a large-scale corpus to build a high-capacity language model, and the next stage is specific fine-tuning to perform downstream tasks, such as natural language generation (NLG) and natural language understanding (NLU).

**NLG Tasks.** We adopt a decoder-only pre-training paradigm similar to GPT to train the model. Specifically, our model utilizes SRWKV and SRFFN modules to process the input token sequence and generate an output distribution for each target token. Formally, given a token sequence $\mathcal{C} = \{c_1, c_2, \cdots, c_n\}$, we use the standard language modeling objective to maximize the following likelihood:

$$\mathcal{P}(c_t) = \text{softmax}(Y'[t]W_e^T) \tag{11}$$

$$\mathcal{L}_p = \sum_{i=1}^{T} log\mathcal{P}(c_i|c_1, c_2, \cdots, c_{i-1}; \Theta) \tag{12}$$

where $W_e^T$ is the token embedding matrix, and $\Theta$ is the set of all model parameters.

After pre-training the model using the loss in Eq. 12, model parameters are fine-tuned to adapt to different downstream tasks in NLG and NLU. For natural language generation tasks, we define a new dataset $\mathcal{D}_G$, where each sample of data consists of a sequence of input tokens. The consistency between the pre-training process and the NLG task allows for the fine-tuning procedure to reuse the method and objectives adopted in the pre-training pipeline (Eq. 11 and Eq. 12), which maximizes the likelihood of the target token based on the previous information of the target position.

**NLU Tasks.** As for NLU tasks, such as sentiment classification, the fine-tuning process requires several modifications to the top-level of the pre-trained SpikeGPT model to adapt to NLU tasks, as shown in Fig. 2. Specifically, given a new dataset for the downstream task $\mathcal{D}_U$, where each instance consists of a token sequence $\mathcal{C}_i$ and label $l_i$, the following objective is maximized:

$$\mathcal{L}_{NLU} = \sum_{(\mathcal{C}_i, l_i)} l_i * logP(\mathcal{C}_i) \tag{13}$$

where $\mathcal{P}(\mathcal{C}_i)$ is defined as:

$$\mathcal{P}(\mathcal{C}_i) = \text{softmax}(YW_m^T) \tag{14}$$

$W_m^T$ includes the learnable parameters of the MLP module in the top layer, and $Y$ is generated by passing the input token sequence $\mathcal{C}_i$ through the model and then average-pooling the embedding of each target, which is formalized as:

$$Y = \text{AvgPooling}(X_L[1], X_L[2], \cdots, X_L[T]) \tag{15}$$

In the inference phase, we directly give a prompt for the NLG task, and let the model continue and calculate bits-per-character (BPC) and perplexity (PPL) as an evaluation metric. For the NLU task, we pass the inputs through our model to obtain the embedding over target positions, and then apply a mean-pooling operation to all embedded tokens to predict the label of each instance.

### 3.7 Theoretical Energy Consumption Analysis

The key attribute of SNNs here is their remarkable energy efficiency due to dynamical sparsity. We provide a theoretical energy consumption analysis of SpikeGPT's constituent components, while evaluating its feasibility for deployment on asynchronous hardware. The block-level theoretical energy consumption analysis is shown in Tab. 1.

**Spiking RWKV.** The SRWKV block comprises two integral components: the Spiking MLP layer, which receives sparse spike inputs, and the full-precision element-wise product. Within the Spiking RWKV, akin to self-attention mechanisms, the spike input denoted as $X$ undergoes projection through MLP layers, resulting in three distinct variables, namely, $R$, $K$, and $V$. Notably, all these MLP layers operate on spike inputs,

facilitating the transformation of matrix multiplication into sparse addition. It is worth highlighting that the non-binary element-wise product, as depicted in Tab. 1, constitutes a relatively minor proportion compared to the MLP layers. Importantly, it is essential to note that all these operations can be efficiently supported by neuromorphic chip technology.

**SRFFN.** The SRFFN block primarily consists of MLP layers and element-wise product operations. However, an alternative operation is introduced when employing the $ReLU^2$ activation function instead of spiking neurons, While $ReLU^2$ activations are not binary, they induce dynamical sparsity. We can further employ quantization techniques to convert them into integers, a common practice in LLMs (Liu et al., 2023; Dettmers et al., 2022b;a) and SNNs (Venkatesh et al., 2024; Shen et al., 2023). where 8-bit activations and weights can be used with no noticeable loss in performance (Xiao et al., 2023). Moreover, we investigated the possibility of using LIF to replace $ReLU^2$ in Sec. 4.4, which demonstrates that the impact of this substitution on performance is negligible. With regard to integer sparse spikes, modern neuromorphic chips (Orchard et al., 2021; Parpart et al., 2023) accommodate graded spikes that support up to 32-bit integer, rendering the entire SRFFN block compatible with neuromorphic hardware, requiring solely addition operations for its execution.

Table 1: Energy Evaluation: $FL_{MLP}$ represents the Floating-Point Operations (FLOPs) of the MLP layers in the ANNs, while $\hat{R}$ denotes the spike firing rates (indicating the proportion of non-zero elements in the spike matrix) within the spike matrices. In computing operation counts, models are parameterized with $T = 3072$, $d = 512$, $\hat{R} = 0.15$, all derived from real test data of SpikeGPT-46M. Additionally, energy consumption is characterized by assuming $E_{MAC} = 4.5$pJ and $E_{AC} = 0.9$pJ based on Horowitz (2014) and Rathi & Roy (2021). Other than element-wise operation involving dot products in SpikeGPT, all other module inputs are in spike form, leading to a significant reduction in operations by a factor of up to $32.2\times$ overall. This is reflected in the Vanilla GPT-to-SpikeGPT (V/S) ratio.

| | | Vanilla GPT (Radford et al., 2018) (with GLU (Dauphin et al., 2017)) | SpikeGPT (**This work**) | Energy Consumption (pJ) Vanilla GPT | SpikeGPT | Ratio V/S |
|---|---|---|---|---|---|---|
| Attention | $Q/R, K, V$ | $E_{MAC} \cdot 3Td^2$ | $E_{AC} \cdot \hat{R} \cdot 3Td^2$ | $1.09 \times 10^{10}$ | $3.25 \times 10^8$ | $33.3\times$ |
| | $f(Q/R, K, V)$ | $E_{MAC} \cdot 2T^2d$ | $E_{MAC} \cdot 7Td$ | $8.50 \times 10^7$ | $4.95 \times 10^7$ | $1.71\times$ |
| | Scale | $E_{MAC} \cdot T^2$ | - | $4.25 \times 10^7$ | - | - |
| | Softmax | $E_{MAC} \cdot 2T^2$ | - | $8.50 \times 10^7$ | - | - |
| FFN | Layer 1 | $E_{MAC} \cdot FL_{MLP1}$ | $E_{AC} \cdot \hat{R} \cdot FL_{MLP1}$ | $3.62 \times 10^9$ | $1.09 \times 10^8$ | $33.3\times$ |
| | Layer 2 | $E_{MAC} \cdot FL_{MLP2}$ | $E_{AC} \cdot \hat{R} \cdot FL_{MLP2}$ | $1.45 \times 10^{10}$ | $4.35 \times 10^8$ | $33.3\times$ |
| | Layer 3 | $E_{MAC} \cdot FL_{MLP3}$ | $E_{AC} \cdot \hat{R} \cdot FL_{MLP3}$ | $3.62 \times 10^9$ | $1.09 \times 10^8$ | $33.3\times$ |
| Overall | - | - | - | $3.29 \times 10^{10}$ | $1.02 \times 10^9$ | $32.2\times$ |

# 4 Experiments

## 4.1 Experiment Settings

We test two variants of the 46 million parameter model; one where $T = 1,024$ and another where $T = 3,072$. We used the Enwik8 dataset to conduct both training and testing in 46M scale, and our most extensive model with 216 million parameters was trained using the OpenWebText2 (Gao et al., 2020) corpus for pre-training. In our 46M model, we employ a character-level tokenizer, as has been done in previous works Zhai et al. (2021). To evaluate the performance of our model, we calculate its BPC metrics. To mitigate the issue of overfitting, we incorporate dropout after the output of each SRFFN block and set the dropout ratio to 0.03. The 216M model with pre-training consists of 18 layers and a feature dimension $d = 768$. We employ the Byte Pair Encoding (BPE) tokenizer and share the same hyper-parameters as GPT-NeoX (Black et al., 2022). Due to the availability of sufficient data for pre-training, we do not incorporate dropout as we did in our 46M model and remove the binary embedding but use the first layer neurons for encoding. To facilitate better convergence, we utilize a warmup technique during the first 500 training steps. For both the 46M and 216M models, we use the Adam optimizer, and set the learning rate to $6 \times 10^{-4}$ and $4 \times 10^{-4}$, respectively.

All experiments were conducted on four NVIDIA V100 graphic cards. For the models of 46M and 216M, we trained them for 12 and 48 hours respectively.

We evaluated SpikeGPT on two major language-related tasks: Natural Language Generation (NLG) and Natural Language Understanding (NLU). For NLG, we evaluated the text generation performance of SpikeGPT on three classic text generation datasets: Enwik8 (Mahoney, 2011), WikiText-2 (Merity et al., 2017), and WikiText-103 (Merity et al., 2017). These datasets are widely used to measure a model's ability to perform compression and for benchmarking. For NLU, we evaluated the performance of SpikeGPT on four classic text classification datasets: MR (Pang & Lee, 2005), SST-5 (Socher et al., 2013), SST-2 (Socher et al., 2013), and Subj (Pang & Lee, 2004). These datasets cover sentiment analysis and subjective/objective classification. Our implementation is based on PyTorch (Paszke et al., 2019) and SpikingJelly (Fang et al., 2020). For detailed information regarding the datasets and baseline methods, please refer to Appendix B. Our zero-shot text samples of this experiment can be found in Appendix E.

## 4.2 Results on Natural Language Generating Tasks

A summary of results are provided in Tabs. 2 and 3. This includes the BPC and PPL achieved on NLG tasks using SpikeGPT trained on Enwik8, WikiText-103, and WikiText-2 compared to several baselines, including 46M and 216M parameter variants.

Table 2: Enwik8 results. Measured in Bits Per Character (BPC): the lower the better. Baseline comparisons are made with Reformer (Kitaev et al., 2020), Synthesizer (Tay et al., 2020), Linear Transformer (Katharopoulos et al., 2020a), Performer (Choromanski et al., 2020), Stacked LSTM (Graves, 2013) and SHA-LSTM (Merity, 2019). $L, d$ and $T$ denote the number of blocks (network depth), dimension of features, and sequence length, respectively. Both Linear Transformer and Performer are implemented with customized CUDA kernels (`github.com/idiap/fast-transformers`), and all other models are implemented in native Pytorch.

| Method | Spiking | $L$ | $d$ | $T$ | Train BPC | Test BPC | Complexity | Params. |
|---|---|---|---|---|---|---|---|---|
| Transformer | ✗ | 12 | 512 | 1024 | 0.977 | 1.137 | $\mathcal{O}(T^2 \cdot d)$ | 43.0M |
| Reformer | ✗ | 12 | 512 | 1024 | 1.040 | 1.195 | $\mathcal{O}(TlogT \cdot d)$ | 40.1M |
| Synthesizer | ✗ | 12 | 512 | 1024 | 0.994 | 1.298 | $\mathcal{O}(T \cdot d^2)$ | 42.8M |
| Linear Transformer | ✗ | 12 | 512 | 1024 | 0.981 | 1.207 | $\mathcal{O}(T \cdot d^2)$ | 43.0M |
| Performer | ✗ | 12 | 512 | 1024 | 1.002 | 1.199 | $\mathcal{O}(T \cdot d^2logd)$ | 43.0M |
| Stacked LSTM | ✗ | 7 | - | - | 1.420 | 1.670 | $\mathcal{O}(T \cdot d^2)$ | - |
| SHA-LSTM (no attention) | ✗ | 4 | 1024 | 1024 | - | 1.330 | $\mathcal{O}(T \cdot d^2)$ | - |
| **SpikeGPT 46M** | ✓ | **12** | **512** | **1024** | **1.113** | **1.283** | $\mathcal{O}(T \cdot d)$ | 46.1M |
| **SpikeGPT 46M** | ✓ | **12** | **512** | **3072** | **0.903** | **1.262** | $\mathcal{O}(T \cdot d)$ | 46.1M |

As shown in Tab. 2, the generative performance of SpikeGPT has far surpassed that of models with an LSTM backbone, and can approach or even surpass some simplified variants of the Transformer, such as Linear Transformer and Synthesizer. However, it should be pointed out that there is still a certain gap between SpikeGPT and the vanilla Transformer. While increasing the size of $T$ significantly reduces the training BPC of SpikeGPT, its test BPC has not changed significantly, indicating that SpikeGPT is potentially suffering from over-fitting.

We also compared the Perplexity of SpikeGPT and GPT-2 based on WikiText-2 and WikiText-103 datasets on text generation tasks. The results are shown in Tab. 3. In the interest of fairly comparing models of similar scales, we selected GPT-2 small and GPT-2 medium (Radford et al., 2019) with parameter sizes similar to those of the fine-tuned 216M SpikeGPT. We found that after fine-tuning, the performance of SpikeGPT on WikiText-2 has surpassed that of GPT-2 series. Unfortunately, the performance of SpikeGPT on the larger WikiText-103 dataset has fallen behind the GPT-2 series models, which suggests a potential need for refined, and perhaps more sophisticated, training methodologies for SpikeGPT when dealing with larger scale corpora, e.g., knowledge distillation (Bal & Sengupta, 2023).

Table 3: Results on WikiText-2 and WikiText-103 measured in token-level perplexity. Lower values indicate better performance. We report the perplexity when the lowest value was achieved on the validation datasets. Note that for WikiText-2, we use both the WikiText-103 and WikiText-2 training sets to extend the training corpus.

| Method | Parameters | WikiText-103 | | WikiText-2 | |
|---|---|---|---|---|---|
| | | Val. | Test. | Val. | Test. |
| GPT-2 Small  (Radford et al., 2019) | 124M | - | 29.41 | - | 37.50 |
| GPT-2 Medium  (Radford et al., 2019) | 346M | - | 26.37 | - | 22.76 |
| **SpikeGPT With Pre-training** | **216M** | **39.92** | **39.75** | **19.17** | **18.01** |

### 4.3   Results on Natural Language Understanding Tasks

For NLU tasks, we utilize SpikeGPT as a dynamic embedding generator that constructs embeddings based on context. We compare SpikeGPT with text classification algorithms of similar scales, including LSTM (Hochreiter & Schmidhuber, 1997), TextCNN (Kim, 2014), BERT (Kenton & Toutanova, 2019), and the latest SNN-based text classification algorithm TextSCNN (Lv et al., 2023b). The accuracy on four datasets is shown in Tab. 4. The fine-tuned 216M SpikeGPT achieves the second-highest performance among the models, only surpassed by BERT. BERT is a bidirectional Transformer encoder that uses masked training to obtain a high-quality text embedding. However, unlike SpikeGPT, BERT does not have the capability to generate text directly. Our 46M model without any fine-tuning also achieves competitive results compared to the baseline models, indicating the potential of SpikeGPT in NLU tasks. We also analyze the complexity of each method and show that SpikeGPT can achieve linear complexity by using spiking neurons and a recurrent structure. Unlike TextSCNN, our model does not require an additional temporal dimension for processing, as it uses the sequence dimension iteratively during the forward-pass through the spiking neurons.

Table 4: Results of NLU tasks on four text classification datasets using both SNN and ANN methods. Measured in classification accuracy: the higher the better. In the 'complexity per layer' column, we compute the complexity of each method using the following notation: $T$ is sequence length, $d$ is the embedding dimension, $K$ is the convolution kernel size, and $T_{additional}$ is the additional temporal dimension for feed-forward processing. Our model combines recurrent and spiking features and does not need an extra temporal dimension for feed-forward processing, as it exploits the inherent temporal dimension in the language model.

| Method | Spiking | Recurrent | Complexity per layer | SST-2 | SST-5 | MR | Subj. |
|---|---|---|---|---|---|---|---|
| TextCNN (Kim, 2014)[EMNLP] | ✗ | ✗ | $\mathcal{O}(K \cdot T \cdot d^2)$ | 81.93 | 44.29 | 75.02 | 92.20 |
| TextSCNN-Direct Training (Lv et al., 2023b)[ICLR-2023] | ✓ | ✗ | $\mathcal{O}(T_{additional} \cdot K \cdot T \cdot d^2)$ | 75.73 | 23.08 | 51.55 | 53.30 |
| TextSCNN-ANN2SNN+Fine-tune (Lv et al., 2023b)[ICLR-2023] | ✓ | ✗ | $\mathcal{O}(T_{additional} \cdot K \cdot T \cdot d^2)$ | 80.91 | 41.63 | 75.45 | 90.60 |
| LSTM (Tai et al., 2015) | ✗ | ✓ | $\mathcal{O}(T \cdot d^2)$ | 84.92 | 46.43 | 81.60 | - |
| BERT (Kenton & Toutanova, 2019) | ✗ | ✗ | $\mathcal{O}(T^2 \cdot d)$ | 91.73 | 53.21 | 86.72 | - |
| **SpikeGPT 46M** | ✓ | ✓ | $\mathcal{O}(T \cdot d)$ | 80.39 | 37.69 | 69.23 | 88.45 |
| **SpikeGPT 216M** | ✓ | ✓ | $\mathcal{O}(T \cdot d)$ | 82.45 | 38.91 | 68.11 | 89.10 |
| **SpikeGPT 216M With Pre-training** | ✓ | ✓ | $\mathcal{O}(T \cdot d)$ | **88.76** | **51.27** | **85.63** | **95.30** |

### 4.4   A Study of SpikeGPT and RWKV Variants

We conducted a study of variants of SpikeGPT and RWKV variants to evaluate how the binarization of activations impacts performance. The following models are tested:

1. Vanilla RWKV: The original RWKV model as reported in  Peng et al. (2023).

2. Heaviside RWKV: The above RWKV model where neuron activations are binarized.

3. SpikeGPT-B: The model above, where the second-MLP activation is swapped from $ReLU^2$ to a spiking neuron LIF.

4. SpikeGPT: The model described and proposed in this paper.

Results for those experiments are shown in Tab. 5. For the Heaviside RWKV variant, we applied the Heaviside function directly to the RWKV layer for binarization, in lieu of the LIF neuron. Specifically, we employed the Heaviside function for the feed-forward process, and use the arctangent surrogate function to address the non-differentiable nature of the Heaviside function during the backward pass, similar to the original SpikeGPT. For SpikeGPT-B, we substituted the middle activation function in the SRFFN layer (as illustrated in Eq. 10) with the spiking neuron LIF to introduce layer-level binary spiking.

The test BPC of SpikeGPT-B is marginally lower (and thus, better) than that of SpikeGPT. This shows that RFFN layers are tolerant to binarized activations, provided that recurrent dynamics are included. In absence of recurrent dynamics, the Heaviside-RWKV experiment shows the worst performance, thus indicating that single-order recurrence is necessary to compensate for the performance degradation of binarized activations.

Table 5: An ablation study showcases the impact of different architectural modifications on the performance of SpikeGPT models, with all experiments conducted under the consistent settings of $L = 12$, $T = 1024$, and $d = 512$. The evaluation includes Vanilla RWKV as the baseline model without any modifications, Heaviside RWKV, which replaces all spiking neurons with Heaviside functions, SpikeGPT-B featuring a variation where the middle activation in the SRFFN block is also a spiking neuron rather than $ReLU^2$, and SpikeGPT, representing the default configuration of SpikeGPT.

| Model | Vanilla RWKV | Heaviside RWKV | SpikeGPT-B | SpikeGPT |
|---|---|---|---|---|
| **Train BPC** | 1.014 | 1.318 | 1.109 | 1.113 |
| **Test BPC** | 1.201 | 1.403 | 1.306 | 1.283 |

## 4.5 Scaling SpikeGPT

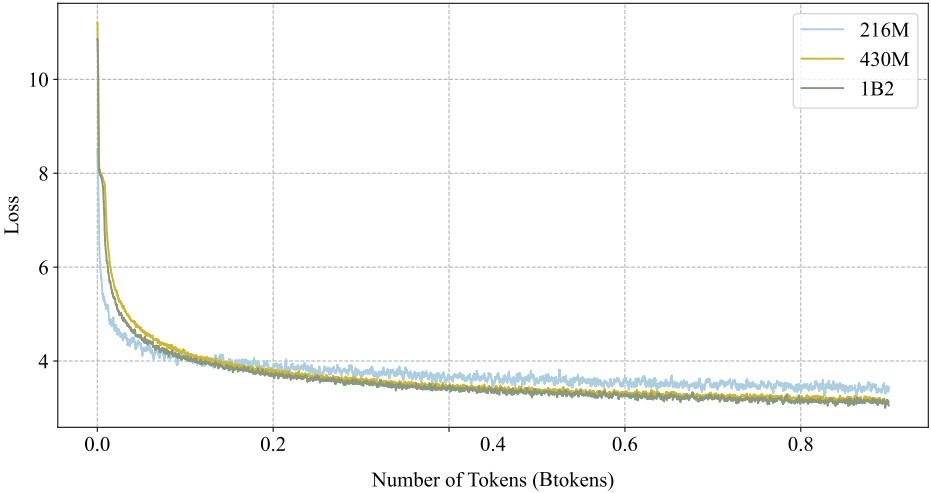

Figure 3: Training Loss for Different Model Sizes on 0.9B Tokens.

Neural scaling laws posit that model error decreases as a power function of training set size and model size, and have given confidence in performance. Such projections become important as training becomes increasingly expensive with larger models. A widely adopted best practice in LLM training is to first test scalability with smaller models, where scaling laws begin to take effect (Kaplan et al., 2020; Hoffmann et al., 2022; Yang et al., 2023; Zhu et al., 2024). The GPT-4 technical report revealed that a prediction model just $1/10,000$ the size of the final model can still accurately forecast the full-sized model performance (Achiam et al., 2023).

We evaluate the scaling capability of SpikeGPT by training models of three different sizes: 216M, 430M, and 1B2, using the MiniPile (Kaddour, 2023) dataset with 0.9B tokens. Due to the constrain on the computational resources, we cannot fully train the 430M and 1B2 model with OpenWebText2. As shown in Fig. 3, the training loss decreases as the model size increases, demonstrating SpikeGPT's ability to scale effectively. The 216M model achieves a final training loss of 3.20, while the 430M model reaches a loss of 3.00, and the 1B2 model attains the lowest loss of 2.88. These results suggest that SpikeGPT follows the neural scaling laws, with larger models exhibiting better performance when trained on the same dataset.

Furthermore, we observe that the rate of improvement in training loss slows down as the model size increases, which is consistent with the diminishing returns predicted by scaling laws. The performance gain from 216M to 430M is more significant than the improvement from 430M to 1B2, indicating that the benefits of increasing model size gradually taper off. This insight can help guide decisions on model size selection, balancing performance gains with computational costs.

## 5 Conclusion

Our results demonstrate that event-driven spiking activations are not only capable of language generation, but they can do so with fewer high-cost operations. We develop techniques that promote lightweight models for the NLP community, and make large-scale models for the neuromorphic and SNN community more effective. We demonstrate how large SNNs can be trained in a way that harnesses advances in transformers and our own serialized version of the attention mechanisms. We expect this research can open new directions for large-scale SNNs.

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

## APPENDIX

## A  Details about LIF Neuron

### A.1  Derivation and Analysis of the LIF Neuron

ANNs and SNNs can model similar network topologies, but SNNs use spiking neurons instead of artificial ones (Rosenblatt, 1958). Spiking neurons, like artificial ones, work on a weighted sum of inputs. Instead of using a sigmoid or ReLU nonlinearity, this sum affects the neuron's membrane potential $U(t)$. When the potential exceeds a threshold $\theta$, the neuron spikes, sending a signal to connected neurons. Most inputs are brief electrical spikes, unlikely to arrive simultaneously, indicating temporal dynamics that sustain the membrane potential.

Louis Lapicque, in 1907, demonstrated that spiking neurons resemble a low-pass filter circuit with a resistor $R$ and capacitor $C$, known as the leaky integrate-and-fire (LIF) neuron (Lapique, 1907; Brunel & Van Rossum, 2007). Physiologically, the capacitance comes from the lipid bilayer membrane, and the resistance from ion channels (Hodgkin & Huxley, 1952). This can be modeled as:

$$\tau \frac{dU(t)}{dt} = -U(t) + I_{\text{in}}(t)R \tag{16}$$

where $\tau = RC$ is the time constant. For a constant input current, the solution is:

$$U(t) = I_{\text{in}}R + [U_0 - I_{\text{in}}R]e^{-\frac{t}{\tau}} \tag{17}$$

This shows exponential relaxation of $U(t)$ to a steady state. For a sequence-based neural network, the forward Euler method approximates the solution:

$$U[t] = \beta U[t-1] + (1-\beta)I_{\text{in}}[t] \tag{18}$$

Here, $\beta = e^{-1/\tau}$ is the decay rate. In deep learning, the input weight is usually a learnable parameter. Simplifying, the input current coefficient becomes a learnable weight $W$, leading to $I_{\text{in}}[t] = WX[t]$. Including spiking and reset, we get:

$$U[t] = \beta U[t-1] + WX[t] - S_{\text{out}}[t-1]\theta \tag{19}$$

$S_{\text{out}}[t] \in \{0,1\}$ is the neuron's spike output. If $S_{\text{out}} = 1$, the reset term subtracts the threshold $\theta$; otherwise, it has no effect. A spike is generated if:

$$S_{\text{out}}[t] = \begin{cases} 1, & \text{if } U[t] > \theta \\ 0, & \text{otherwise} \end{cases} \tag{20}$$

### A.2  Backpropagation through LIF Neuron

Here we describe how backpropagation can be applied to LIF neurons. At time-step $t$, $\boldsymbol{H}_t^i$ and $\boldsymbol{U}_t^i$ represent the membrane potential after neuronal dynamics and after reset, respectively. The vectors $\boldsymbol{U}_{th}^i$ and $\boldsymbol{U}_{reset}^i$ indicate the threshold and reset potential, respectively. The weighted inputs from the preceding layer are given by $\boldsymbol{X}_t^i = \boldsymbol{W}^{i-1}\boldsymbol{I}_t^i$. The output spike at time-step $t$ is represented by $\boldsymbol{S}_t^i = [s_{t,j}^i]$, where $s_{t,j}^i = 1$ if the $j$-th neuron fires a spike, otherwise $s_{t,j}^i = 0$. The gradients propagating back from the next layer are $\frac{\partial L_t}{\partial \boldsymbol{S}_t^i}$. We

can recursively compute the gradients as follows:

$$\frac{\partial L}{\partial \boldsymbol{H}_t^i} = \frac{\partial L}{\partial \boldsymbol{H}_{t+1}^i}\frac{\partial \boldsymbol{H}_{t+1}^i}{\partial \boldsymbol{H}_t^i} + \frac{\partial L_t}{\partial \boldsymbol{H}_t^i} \tag{21}$$

$$\frac{\partial \boldsymbol{H}_{t+1}^i}{\partial \boldsymbol{H}_t^i} = \frac{\partial \boldsymbol{H}_{t+1}^i}{\partial \boldsymbol{U}_t^i}\frac{\partial \boldsymbol{U}_t^i}{\partial \boldsymbol{H}_t^i} \tag{22}$$

$$\frac{\partial L_t}{\partial \boldsymbol{H}_t^i} = \frac{\partial L_t}{\partial \boldsymbol{S}_t^i}\frac{\partial \boldsymbol{S}_t^i}{\partial \boldsymbol{H}_t^i} \tag{23}$$

Using Eq. 1, we obtain:

$$\frac{\partial \boldsymbol{H}_{t+1}^i}{\partial \boldsymbol{U}_t^i} = 1 - \beta \tag{24}$$

$$\frac{\partial \boldsymbol{U}_t^i}{\partial \boldsymbol{H}_t^i} = 1 - \boldsymbol{S}_t + (\boldsymbol{U}_{reset}^i - \boldsymbol{H}_t^i)\frac{\partial \boldsymbol{S}_t^i}{\partial \boldsymbol{H}_t^i} \tag{25}$$

$$\frac{\partial \boldsymbol{S}_t^i}{\partial \boldsymbol{H}_t^i} = \Theta'(\boldsymbol{H}_t^i - \boldsymbol{U}_{th}^i) \tag{26}$$

$$\frac{\partial \boldsymbol{H}_t^i}{\partial \boldsymbol{X}_t^i} = \beta \tag{27}$$

Note that $\frac{\partial *}{\partial \boldsymbol{S}_t^i} = 0$ when $t \geq T$, and $\boldsymbol{U}_{-1}^i = \boldsymbol{U}_{reset}^i$. We use the derivative of the surrogate function $\sigma(x)$ to define the derivative of the Heaviside function $\Theta(x)$. For more details, please refer to Interactive tutorial notebooks on snntorch[1].

## B  Datasets and Baselines

### B.1  Datasets

We conducted experiments on two major types of tasks, Natural Language Generation (NLG) and Natural Language Understanding (NLU).
For NLG tasks, we chose the following 3 classic text classification datasets to evaluate the text generation performance of SpikeGPT: Enwik8 (Mahoney, 2011), WikiText-2 (Merity et al., 2017) and WikiText-103 (Merity et al., 2017).

- Enwik8. The Enwik8 dataset is a subset of the English Wikipedia XML dump from March 2006. It contains the first 100 million bytes of the dump and is typically used to measure a model's ability to compress data. The dataset is based on the Hutter Prize, a competition for lossless compression of human knowledge. We split the tokens into three subsets: 90% for training, 5% for validation, and 5% for testing.

- WikiText-2. WikiText-2 is a natural language dataset comprising a collection of 2 million tokens derived from Wikipedia articles. This dataset is commonly utilized for benchmarking various natural language processing models.

- WikiText-103. The Wikitext-103 dataset is a large collection of text extracted from Wikipedia articles that are verified as Good or Featured. It contains over 100 million tokens and covers a wide range of topics and domains. The dataset is suitable for language modeling tasks that require long-term dependencies and rich vocabulary. The Wikitext-103 dataset is a larger and more diverse version of the Wikitext-2 dataset.

---

[1]https://snntorch.readthedocs.io/en/latest/tutorials/index.html

For NLU tasks, we chose the following 4 classic text classification datasets to evaluate the performance of our proposed SpikeGPT: MR (Pang & Lee, 2005), SST-5 (Socher et al., 2013), SST-2 (Socher et al., 2013), Subj. (Pang & Lee, 2004)

- MR (Pang & Lee, 2005). It consists of movie review files, labeled based on their overall sentiment polarity (positive or negative) or subjective rating.

- SST-5. The Stanford Sentiment Tree Library 5 includes 11855 sentences extracted from movie reviews for sentiment classification (Socher et al., 2013). There are 5 different categories (very negative, negative, neutral, positive, and very positive)

- SST-2 (Socher et al., 2013). It is a binary version of SST-5, with only two classes (positive and negative).

- Subj (Pang & Lee, 2004). Classify sentences in the dataset as subjective or objective.

The sample sizes and text lengths of these datasets vary. If there is no standard training test segmentation, we will follow Lv et al. (2023b) and randomly select 10% of the samples from the entire dataset as the test set.

### B.2 Baselines

To verify the effectiveness on NLG and NLU tasks of our proposed SpikeGPT, we compare it with the following representative baselines:

For NLG, we list the baselines that we have selected as follows:

- **Stacked LSTM**. A model architecture that stacks multiple LSTM modules together.

- **SHA-LSTM (Merity, 2019)**. An LSTM model that follows by a single attention head layer.

- **Transformer (Vaswani et al., 2017)**. Transformer is a state-of-the-art neural network architecture, leveraging self-attention mechanisms to capture global dependencies of sequential data.

- **Reformer (Kitaev et al., 2020)**. Reformer is an extensible variant of the Transformer model. By introducing the invertible sheaf and using the local sensitive hash mechanism, it solves the problem of low memory and computing efficiency of the traditional Transformer, and realizes efficient processing of long sequences.

- **Synthesizer (Tay et al., 2020)**. Synthesizer is also a variant of Transformer, which is a model that learns to synthesize attention weights without token-token interaction.

- **Linear Transformer (Katharopoulos et al., 2020b)**. Linear Transformer is a lightweight variant of Transformer that uses linear transformation layers to construct a self attention mechanism.

- **Performer (Choromanski et al., 2020)**. A variant of Transformer that does not depend on sparsity or low-rankness assumptions and could use linear complexity to accurately estimate attention weights.

- **GPT-2 (Radford et al., 2019)**. GPT-2 is a transformer-based language model that specifically functions as a decoder. It is an extensively trained, large-scale generative model using the autoregressive paradigm. To ensure compatibility with the parameter sizes of SpikeGPT, we selected GPT-2 medium and GPT-2 small as suitable alternatives.

For NLU, the baselines we have selected are as follows:

- **LSTM (Hochreiter & Schmidhuber, 1997)**. LSTM model is a type of recurrent neural network with the ability to capture and utilize long-term dependencies in input sequences.

- **TextCNN (Kim, 2014)**. TextCNN is a convolutional neural network architecture specifically designed for text classification tasks, leveraging convolutional layers to capture local patterns and features in textual data.

- **TextSCNN (Lv et al., 2023b)**. A variant of TextCNN model that combines spiking neural networks.

- **BERT (Kenton & Toutanova, 2019)**. BERT is a bidirectional language model based on the Transformer Encoder-only architecture and an auto-encoding training paradigm.

## C  Details of RWKV

### C.1  Token Shift

Given an input $X$, we perform a *token shift* operation on it as follows:

$$\begin{aligned}
X_s &= \text{ZeroPad}_{[0,0,-1,1]}(X) \\
W_{\text{shift}} &= \left[ (\frac{i}{E})^{n/N} \right], i = 1, \cdots, E \\
\mathcal{X} &= W_{\text{shift}} \odot X + (1 - W_{\text{shift}}) \odot X_s
\end{aligned} \tag{28}$$

where ZeroPad[2] denotes the zero padding operation, $W_{\text{shift}}$ represents a learnable shift mask, $E$ is the embedding size of each token, $t$ is the current block, and $T$ is the total number of blocks.

### C.2  General RWKV

Inspired by the Attention Free Transformer (Zhai et al., 2021), RWKV acts as a replacement for self-attention. It reduces computational complexity by swapping matrix-matrix multiplication with a convolution that sweeps along the time dimension. We subsequently modify this step to instead operate recurrently on input data. This modification enables compatibility with recurrent SNNs, thus making it more manageable to run on limited resources.

Given an input token-shifted embedding vector $\mathcal{X}$, similar to self-attention, RWKV first applies a linear transform $R = \mathcal{X} M_R, K = \mathcal{X} M_K, V = \mathcal{X} M_V$[3]. $\mathcal{X}$ is a time-varying embedding (varying over the sequence), and so $R, K, V$ are also time-varying. Fig. 1 depicts the sequence unrolled into a set of 2-D matrices.

$M_R$, $M_K$ and $M_V$ consist of learnable parameters, where $K$ and $V$ can be likened to the key and value matrices of self-attention. $R$ is referred to as the receptance matrix, where each element indicates the acceptance of past information.

Next, the following operation is applied:

$$Y_t = \sigma(R_t) \odot \frac{\sum_{i=1}^{t} \exp(W_{(T+i-t)}) \odot \exp(K_i) \odot V_i}{\sum_{i=1}^{t} \exp(W_{(T+i-t)}) \odot \exp(K_i)} \tag{29}$$

where $\odot$ is the element-wise product, $T$ is the sequence length, $\sigma$ is the non-linearity applied to $R$ with the default being Sigmoid; $W \in \mathbb{R}^{T \times E}$ is the positional weight decay matrix (represented as a vector unrolled over time in Fig. 1). $W$ encodes the sequential importance of a given word on subsequent words. It is not directly learnable, but it varies over time with learnable dynamics. Long-range dependence can be captured when the dynamics are learnt to decay slowly. The parallel version of RWKV is also shown in Fig. 5.

Intuitively, as time $t$ increases, the vector $Y_t$ is dependent on a longer history, represented by the summation of an increasing number of terms.

---

[2]The subscript $[0, 0, -1, 1]$ is written with PyTorch syntax in mind, where $-1$ clips the top row and $1$ zero-pads the bottom row.

[3]$\{M_R, M_K, M_V\} \in \mathbb{R}^{E \times H}$, where $H$ denotes hidden size. In RWKV, we set $E = H$.

For the target position $t$, RWKV performs a weighted summation in the positional interval of $[1, t]$, and takes the Hadamard product of the weighted result with the receptance $\sigma(R_t)$. By taking the Sigmoid of $R_t$, the receptance acts as a 'forget gate' by eliminating unnecessary historical information.

## C.3  Self-Attention and RWKV

Distinct from the method of calculating the matching degree[4] between tokens by the self-attention mechanism, RWKV decomposes the calculation of matching degree into: $\alpha_{ij} = \sigma(R_i) \odot \exp(W_{T-i+1}) \odot \exp(K_j)$, where $\alpha_{ij} \in \mathbb{R}^E$ is a vector. Each element in $\alpha_{ij}$, that is $\alpha_{ijk}$, represents the matching degree at the k-th position of the embedding of the i-th and j-th tokens. In other words, it can be seen as a multi-headed RWKV with $E$ heads, each of which has a hidden size=1, which is similar to the multi-headed self-attention (MHA) mechanism.

To gain a more comprehensive understanding of SpikeGPT, we conducted visualizations of the spike and membrane potential patterns in the Spiking RWKV layer and Spiking Receptance Feed-Forward Networks (SRFFN) layers (Fig. 4). These visualizations clearly reveal distinct differences between the Spiking RWKV and SRFFN layers, indicating diverse information representation patterns. Notably, the SRFFN layer exhibits a higher firing rate, suggesting that it may retain more information similar to Transformer FFN layers (Geva et al., 2021). It is worth noting that in various studies, outliers in Transformer-based language models have been shown to significantly impact performance, making them a crucial concern in the quantification of large language models (Wei et al., 2022). Therefore, understanding and addressing the implications of outliers in SpikeGPT are of utmost importance, particularly in the context of optimizing its overall performance and reliability. However, due to the binary nature of SNNs, these outliers cannot be expressed in activation values as in ANNs. Nevertheless, a notable observation is the presence of prominent outliers in the membrane potential of individual neurons, many of which are negative. This finding suggests that SpikeGPT employs a different approach to accommodate and preserve outliers.

## C.4  Positional Weight Decay

The positional weight bias matrix $W$ is determined by three matrices, $W_d$, $W_c$ and $W_f$, parameterized as follows:

$$W_d = \ln(W_s), W_s \in \mathbb{R}^{E \times 1} \tag{30}$$

$$W_c = \begin{bmatrix} (-T+2) & (-T+3) & (-T+4) & \cdots & -1 & 0 \end{bmatrix} \in \mathbb{R}^{1 \times (T-1)} \tag{31}$$

$$W_f = \begin{bmatrix} \ln(p_k) & \ln(p_k) & \cdots & \ln(p_k) \end{bmatrix} \in \mathbb{R}^{E \times 1} \tag{32}$$

where $W_s$ is a pre-calculated matrix dependent on the layer and size of $E$, the vector $W_d$ contains the decay factors for each time-step, $W_c$ is an indicator of time-step, and $W_f$ is the initial decay factor to avoid the constant decay phenomenon of RNNs. $W_d$ and $W_f$ are both learnable, and $W_c$ is a static, pre-calculated matrix based on training time-step. $p_k$ is a hyperparameter, which is set to 0.3 in this paper.

## D  Visualization of Spike and Membrane Potential

To gain a more comprehensive understanding of SpikeGPT, we conducted visualizations of the spike and membrane potential patterns in the Spiking RWKV layer and Spiking Receptance Feed-Forward Networks (SRFFN) layers (Fig. 4). These visualizations clearly reveal distinct differences between the Spiking RWKV and SRFFN layers, indicating diverse information representation patterns. Notably, the SRFFN layer exhibits a higher firing rate, suggesting that it may retain more information similar to Transformer FFN layers (Geva et al., 2021). It is worth noting that in various studies, outliers in Transformer-based language models have been shown to significantly impact performance, making them a crucial concern in the quantification of large

---

[4]A scalar in self-attention, $\alpha_{ij} = Q_i K_j^T$

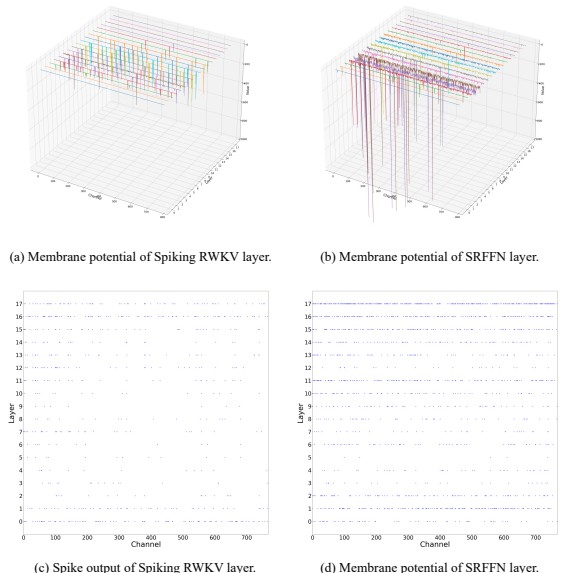

(a) Membrane potential of Spiking RWKV layer. (b) Membrane potential of SRFFN layer.

(c) Spike output of Spiking RWKV layer. (d) Membrane potential of SRFFN layer.

Figure 4: Visualization of spike and membrane potential of neurons. Figure (a) and (b) depict the membrane potential of the Spiking RWKV layer, while figure (c) and (d) display the spike patterns observed in the SRFFN layer, where each dot represents a spike event.

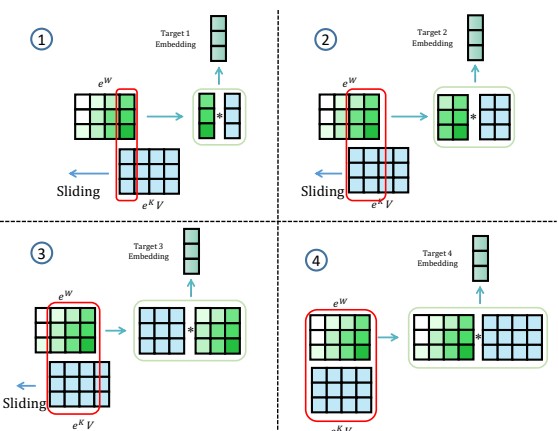

Figure 5: A demonstration of the Parallelized RWKV model adeptly handling computations involving $e^W$ and $e^K V$ through the application of a large-kernel convolution operation. Notably, during the convolution's sliding window process, the model implements a decay mechanism to effectively manage temporal dependencies. For this demonstration, the sequence length is set to $N = 4$, and the embedding size is configured to $E = 3$.

language models (Wei et al., 2022). Therefore, understanding and addressing the implications of outliers in SpikeGPT are of utmost importance, particularly in the context of optimizing its overall performance and reliability. However, due to the binary nature of SNNs, these outliers cannot be expressed in activation values as in ANNs. Nevertheless, a notable observation is the presence of prominent outliers in the membrane potential of individual neurons, many of which are negative. This finding suggests that SpikeGPT employs a different approach to accommodate and preserve outliers.

## E   Generation Examples

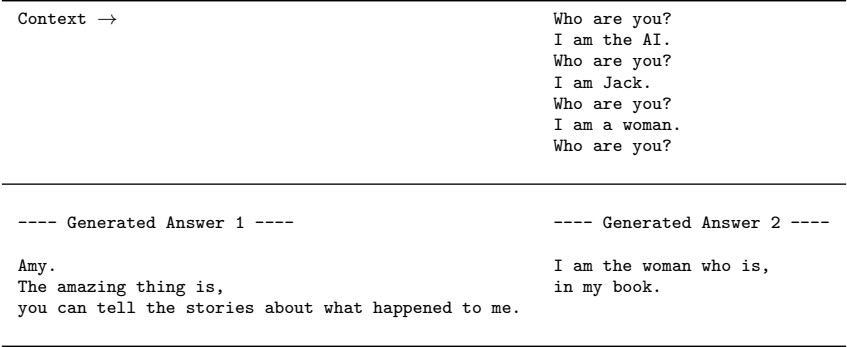

Figure 6: Example of text generated by SpikeGPT. The model is trained on OpenWebText2.

```
Context →                                                                    <start>
```

```
---- Generated Answer ----
"so what do you think of me?   "
I asked.
he was standing behind me, still looking at the posters,
and I suddenly knew what he was asking.
" what do you think of me?   "
I looked up at him , the corners of his lips pressed together.
"I don't know."
he looked at me and I was staring at him, not even realizing what he was doing.
all I could do was watch him,
look at him, listen to his voice,
and then I would fall back to the stage and look at him.
```

Figure 7: Example of text generated by SpikeGPT. The model is trained on BookCorpus.

```
Context →
In a shocking finding, scientist discovered a herd of dragons living in a remote, previously unexplored valley,
in Tibet.  Even more surprising to the researchers was the fact that the dragons spoke perfect Chinese.
```

```
---- Generated Answer ----

In a separate case, they found that their earliest ancestor were extinct.  They did not have the physical
condition of the tree.  The discovery of a Chinese zoo was the latest study by archaeologist Peter Horowitz,
who led the researchers at the University of Texas, who made the discovery a great example.  The researchers
also examined the significance of the artifacts of the bones, the humanoid skeleton, the skull, the cell parts
and the outer jaws.  The species is believed to be located in the mountains and to provide information about the
artifacts of the island.  The discovery shows that this is a large area in the surrounding area of a local island,
which has a huge influx of the populations.
```

Figure 8: Example of text generated by SpikeGPT. The model is trained on OpenWebText2.

```
Context →
Pinky
The pink ghost's AI is designed to "feel" opposite of the red ghost's behavior.  Pinky actually attempts to get
out in front of Pac-Man.  This is accomplished by setting the target 4 tiles ahead of Pac-Man's current location
in the direction that Pac-Man is travelling.  One exception to this is when Pac-Man is traveling up.  Due to an
overflow bug in the code, the calculation includes a left offset equal to the expected up offset.
```

```
---- Generated Answer ----
This is where the AI is able to see how much the AI is still in motion.  This is because the player can control
the location of the player in the real world.  It can also predict the movement of the player by moving the
screen forward towards the player.  This is why the game is so powerful when it comes to setting the target up.
This means that the player can be able to see what is happening in the world, but not exactly where the player is.
This allows the player to see where they are at and how far the player is heading.  The game can also be used
to track the location of the player, and even track the location of the player.  In addition to this, the game
can also track the location of enemies in the sky and detect targets.  It can also also monitor the location of
enemies in the area, such as the locations of buildings, schools, schools, or even buildings where a player is in
a situation where the player can see the world around them.
One of the most interesting features of the game is the ability to control the direction of the player.  This
allows players to choose between the location of the player they want to fly, or the distance of the enemy
location.  This allows players to see the direction of the player, as well as where they want to go.  The game
also supports navigation, and can also track the location of vehicles in the sky.
```

Figure 9: Example of text generated by SpikeGPT. The model is trained on OpenWebText2.

---

Context →
Aaron loves mint chocolate cake, but he requires that it be paired with mini chocolate chips, so I threw some
of those in between the layers.  I also had a few Peppermint Jo Jos on hand so I crushed them up and threw
some of those in along with some crushed meringue cookies because, why not?  It's a total smorgasbord of minty
chocolate chippy cookie crunchy goodness.  I didn't measure how much of each topping I used, but after I tasted
the finished product, I wish I had added more.  You can add anything you want- crushed candy canes, peppermint
bark, etc.  And don't be afraid to use a heavy hand.  Texture = good.

---

---- Generated Answer ----
Pour out of the cupcake, cut out some vanilla butter, then add sugar and vanilla.  Remove from heat until thick
and smooth, then place into the pan for 10 minutes until golden brown.  Turn on and off, stirring until the
cake's browned, as if it were black chocolate and orange syrup.

---

Figure 10: Example of text generated by SpikeGPT. The model is trained on OpenWebText2.

---

Context →
Corporal Michael P. Goeldin was an unskilled laborer from Ireland when he enlisted in Company A in November 1860.
Goldein survived the war.  Corporal Patrick O'Neal, also from Ireland, first enlisted in 1854 and served with
Company L, 3d U.S. Artillery, in Oregon.  He returned to the East Coast and enlisted in the company in 1860.
O'Neal served until 1874, when he was named superintendent of the National Cemetery at Willets Point, New York.
Corporal Benjamin Browne was a shoemaker from Orange County, New York.  In August 1862, he enlisted in the
newly formed 124th New York Volunteers, and was one of sixty-one men who transferred into Company A that October.
Browne reenlisted in the company in February 1864 while it was camped at Brandy Station.  He returned to civilian
life after completing his enlistment in 1867.  On 10 June, Artificer William Collins was promoted to corporal,
probably to fill a combat leadership void for the crossing of the James River.  Collins's service record does not
reflect the qualities he demonstrated to earn this promotion, but he had obviously overcome some serious problems.
Born in Sacketts Harbor, New York, Collins enlisted in the company in December 1853 at the age of twenty-two, and
reenlisted in December 1858.  Just a month before the war began in April 1861, Collins went "over the hill" and
was not caught until three years later.  Returned to the company on 22 March 1864,

---

---- Generated Answer ----

he was tried for possession of marijuana and attempted first degree murder.
For more information on Collins on Twitter visit www.facebook.com/JoeCollins
In April 1879, Collins was sentenced to life in prison for two counts of first degree murder:  first degree first
degree first degree murder, second degree second degree second degree murder,

---

Figure 11: Example of text generated by SpikeGPT. The model is trained on OpenWebText2.

---

Context →
Prehistoric man sketched an incredible array of prehistoric beasts on the rough limestone walls of a cave in
modern day France 36,000 years ago.  Now, with the help of cutting-edge technology, those works of art in the
Chauvet-Pont-d'Arc Cave have been reproduced to create the biggest replica cave in the world.  The manmade
cavern named the Caverne du Pont-d'Arc has been built a few miles from the original site in Vallon-Pont-D'arc
in Southern France and contains 1,000 painstakingly-reproduced drawings as well as around 450 bones and other
features...  Cavemen and women sketched an incredible array of prehistoric beasts on the rough limestone walls of
a cave 36,000 years ago and now a replica has been created (pictured) that show how beautiful the cave was in the
70s and 80s

---

---- Generated Answer ----

Scroll down for video
Growth of a rare cave on a cliff near the river level near the Great Wall of Barcelona (pictured right):  The
world's oldest stone cave in Italy has been discovered in part because of ancient cave erosion.
It has become a key part of modern archaeology to determine how the pyramids lived before the Stone Age.
In an open letter dated 19 June, archaeologists from the University of Barcelona say they found the excavation
and excavation on the site 'significant'.

---

Figure 12: Example of text generated by SpikeGPT. The model is trained on OpenWebText2.

