# OpenReview forum: "SpikeGPT: Generative Pre-trained Language Model with Spiking Neural Networks"
_TMLR — Accepted by TMLR_

### Review · Reviewer_zsxD · 2024-05-08

**Summary Of Contributions:**

This paper proposes SpikeGPT, the first GPT implementation with the Spiking Neural Network architecture. To achieve its goal, the paper resolves the issue of language encoding for SNN, training large-scale SNN, and the incompatible of SNN with self-attention. SpikeGPT leverage the architecture of the RWKV language model, with novel techniques resolving the encoding and training issues. The final outcome leads to an implementation of a 46M and a 216M model, achieving competitive benchmark results with non-spiking models while maintaining 32.2x fewer operations.

**Audience:**

Yes

**Broader Impact Concerns:**

No concern on broader impact

**Claims And Evidence:**

Yes

**Requested Changes:**

See weakness part.

**Strengths And Weaknesses:**

## Strength

This paper is overall well written, well motivated, and technically sound. Being the first paper to achieve GPT-level model on SNN and achieving competitive benchmark results, the paper would have significant impact to the field of SNN and general efficienct deep learning algorithms and systems. Overall a very solid work.

## Weakness

The paper can be further improved by discussing the following topic:
1. As the first paper to present a SNN implementation of GPT, this paper would have a broad impact in the TMLR audiance. However, this paper lacks discussion on fundamental related work of SNN and the discussion of basic SNN operation. Adding a subsection in related work and add a few diagrams/equations to explain SNN operation will make the paper more accessible to broader audiences.
2. It would be interesting to see if the proposed SpikeGPT can benefit from migrating the weights of pretrained non-spiking LLM or distill from a pretrained model, rather than doing the full pretraining from scratch

---

> ### Author Response · Authors · 2024-05-28
> **Author Response**
>
> Dear Reviewer zsxD,
>
> Thank you for your insightful and supportive review of our manuscript. We are delighted to receive such positive feedback, and we appreciate your recognition of the potential impact our work may have on the fields of SNNs and efficient deep learning algorithms and systems. Below, we provide our responses to each of your comments and outline the specific changes:
>
> **As the first paper to present an SNN implementation of GPT, this paper would have a broad impact on the TMLR audience. However, this paper lacks discussion on fundamental related work of SNN and the discussion of basic SNN operation. Adding a subsection in the related work and adding a few diagrams/equations to explain SNN operation will make the paper more accessible to broader audiences.**
>
> Response: This comment is also reflected by Reviewer odWR, and so have given it significant weight. We have provided several resources in the manuscript for readers new to spiking neural networks. we have updated Appendix A with additional background material. This includes a detailed explanation of backpropagation through the LIF neuron and how surrogate gradient descent is employed in training SNNs. Furthermore, we have included the derivation of the LIF neuron using the forward Euler method to provide a solid mathematical foundation. We have also provided a link to interactive tutorial code that can act as a good fundamental resource to SNNs.
>
> **It would be interesting to see if the proposed SpikeGPT can benefit from migrating the weights of pretrained non-spiking LLM or distilling from a pretrained model, rather than doing the full pretraining from scratch.**
>
> Response: In developing SpikeGPT, we tested a variety of models, including those trained from scratch and those pretrained using non-spiking LLMs. Unfortunately, the models pretrained from ANNs performed poorly, as shown in the table below:
>
> | Model                                | Loss |
> | ------------------------------------ | ---- |
> | SpikeGPT 216M (Trained from scratch) | 3.22 |
> | SpikeGPT 216M (Pretrained from ANN)  | 4.57 |

---

> > ### Comment · Reviewer_zsxD · 2024-05-28
> >
> > I would like to thank the authors for the responses. I have no further questions about this paper.

---

### Review · Reviewer_odWR · 2024-05-12

**Summary Of Contributions:**

The authors introduced SpikeGPT, an energy-efficient generative language model employing Spiking Neural Networks (SNNs). This model integrates modifications to the traditional transformer architecture, reducing computational complexity from quadratic to linear by adapting a streaming input mechanism suitable for SNNs. The authors applied innovative techniques tailored for SNN training, including aligning the language sequence with SNN temporal dynamics, autoregressive training that accumulates spikes, and utilizing stateful neurons to overcome binary activation constraints. Experimental results show that SpikeGPT can achieve competitive performance against similar-size transformer models with less energy consumption.

**Audience:**

Yes

**Claims And Evidence:**

Yes

**Requested Changes:**

- To mitigate the steep learning curve associated with training SNNs, the authors could provide detailed documentation, tutorials, and open-source code. This would include clear explanations of the surrogate gradient approach and its implementation. Such resources can help demystify the process, making it more accessible to researchers without a deep background in neuromorphic computing.

- The paper could include a more thorough analysis of the scalability challenges encountered when increasing the size of SNNs. This should cover both theoretical and practical constraints, such as memory usage, processing speed, and the efficiency of surrogate gradients at scale.

**Strengths And Weaknesses:**

**Strengths**:
- The paper successfully integrates spiking neural networks with the transformer model architecture, a novel approach in the field of natural language processing. This integration allows the model to leverage the energy efficiency of SNNs while handling complex language generation tasks.

- By modifying the transformer architecture to replace multi-head self-attention with a mechanism that supports streaming inputs, the authors reduce the computational complexity from quadratic to linear, which is important for scaling up models both in terms of size and efficiency.

- The paper demonstrates that SpikeGPT can achieve substantial reductions in energy consumption on neuromorphic hardware. In the meanwhile,  SpikeGPT remains competitive with conventional non-spiking models on tested benchmarks compared with transformer models.

**Weaknesses**:
- Even though the paper presents a successful implementation, training SNNs is inherently complex due to their non-differentiable nature and the use of surrogate gradients. This complexity might limit wider adoption or replicability by other researchers without substantial expertise in neuromorphic computing.

- While the model scales up to 216 million parameters, the challenges associated with scaling SNNs even further (to the size of the largest ANNs in use today, e.g., billions of parameters with transformer models) are not thoroughly discussed. Potential bottlenecks and computational constraints could be significant as the model size increases.

---

> ### Author Response · Authors · 2024-05-28
> **Author Response**
>
> Dear Reviewer odWR,
>
> We sincerely appreciate your thorough and thoughtful review of our manuscript. Your insightful comments and constructive suggestions demonstrate a deep understanding of our work and its potential impact on the field of natural language processing and neuromorphic computing. Below, we provide our detailed responses and outline the revisions we plan to make in light of your suggestions:
>
> **To mitigate the steep learning curve associated with training SNNs, the authors could provide detailed documentation, tutorials, and open-source code. This would include clear explanations of the surrogate gradient approach and its implementation. Such resources can help demystify the process, making it more accessible to researchers without a deep background in neuromorphic computing.**
>
> Response: We have provided several resources in the manuscript for readers new to spiking neural networks. Appendix A has been updated with additional background material on backpropagation through the LIF neuron and how the surrogate gradient works. We have also provided a link to interactive tutorial code.
>
> **The paper could include a more thorough analysis of the scalability challenges encountered when increasing the size of SNNs. This should cover both theoretical and practical constraints, such as memory usage, processing speed, and the efficiency of surrogate gradients at scale.**
>
> Response: We have added a new subsection titled "Scaling SpikeGPT" which provides insights into our experiments on larger-scale variants of SpikeGPT, demonstrating its scaling performance. Please refer to the blue highlighted part of Section 4.

---

### Review · Reviewer_KfdC · 2024-05-16

**Summary Of Contributions:**

This work presents efficient training of Spike Neural Network (SNN)-based transformers on natural language tasks, obtained by adapting the RWKV architecture to SNN. This is achieved primarily by replacing the multi-head self-attention module with a Spiking RWKV module, which achieves linear complexity with sequence length. They perform autoregressive training on sizeable SSNs (up to 216M parameters) demonstrating reasonable performance against non-spiking networks of similar size on some Natural Language Generation and Natural Language Understanding tasks.

**Audience:**

Yes

**Broader Impact Concerns:**

Not addressed in the manuscript. No concerns on my end.

**Claims And Evidence:**

Yes

**Requested Changes:**

Requested clarifications and polishing as per "Weaknesses". Consider expanding Sec 5 or moving it to the appendix.

**Strengths And Weaknesses:**

STRENGTHS:
- enabling efficient training and scaling of SNN is a topic of clear interest to the audience of this journal
- first demonstration of adapting the RWKV architecture for SNN training
- comparable results (typically moderately behind) in terms of Bits Per Characters, perplexity, and accuracy against non-spiking NN of similar size on a selection of downstream tasks
- presented training and inference workflows for both NLG and NLU tasks, adapted for SNN
- theoretical analysis of energy consumption complements well the aforementioned results, and, as expected, is highly favorable towards SNN (estimate of 32.2x vs. comparable non-spiking NN)

WEAKNESSES:
- what drove the architectural choice of settling on a 46M and a 216M variants? Also, I may have missed it but I couldn't find indicated the number of layers and hidden size dimension for the 216M model
- Sec 3.7 mentions the option to "further employ quantization techniques to convert them [the activations] to integer" but it is not clear whether this is actually done in this study, and what implementation is considered by the energy consumption analysis. It should be noted that the references cited here all refer to non-spiking NN works
- Sec 4.4 "Ablation studies" can be confusing. It compares two SNN variants to a non-spiking RWKV and an unmodified spiking RWKV. The latter is called SpikeGPT-R in this section only, so it wasn't clear this was the configuration used in the rest of the paper
- Sec 4.4 also states "Notably, SpikeGPT-S exhibits superior training loss compared to SpikeGPT-R, indicating that
spiking neurons can enhance model performance in certain cases." when comparing network with LIF in SRFFN (SpikeGPT-S) instead of ReLU2 (SpikeGPT-R) but training performance are at best comparable (1.109 vs. 1.113), especially as no dispersion over multiple seeds is reported, and test performance of SpikeGPT-S are inferior. So this statement appears overblown. Also calling this an "extensive ablation" study is not appropriate.
- The whole Sec 5 "Discussion" reads as an appendix and the authors may want to consider moving it there:
    - Sec 5.1 is superficial, just noting that the membrane potential appears different in SRWKV and SRFNN layers, and that it results in SRFNN having higher firing rate. It's an interesting observation but it isn't really discussed.
    - Sec 5.1 is also repeated verbatim in its entirety with appendix B.3
    - Sec 5.2 consists of one paragraph discussing parallelization efforts of RWKV. Again, interesting (although very limited in scope) but not a discussion.

OTHER MINOR OBSERVATIONS:
- Fig 3 presents SRFNN to the left, followed by SRWKV on the right. It's slightly misleading as the SRFNN blocks follows SRWKV in the SpikeGPT architecture
- Sec 5.2 refers to equation 18 which is not part of the main text but is in an appendix
- typo in Sec 3.3: "binary spikes, This allows" -> "binary spikes. This allows"

---

> ### Author Response · Authors · 2024-05-28
> **Author Response: Part I**
>
> Dear Reviewer KfdC,
>
> Thank you for your comprehensive and insightful review of our manuscript. We greatly appreciate the time and effort you have invested in providing such detailed feedback, which will undoubtedly help us improve the quality and clarity of our work. We have carefully considered your comments and suggestions and are committed to addressing each point to improve the clarity and robustness of our work. Below, we provide a detailed response to your feedback:
>
>  **What drove the architectural choice of settling on 46M and 216M variants? Also, I may have missed it but I couldn't find the number of layers and hidden size dimension indicated for the 216M model.**
>
> Response:  The model sizes were chosen based on two factors:
> 1) The 46M architecture matches the size of the previous largest SNN, with the same number of layers and hidden neurons as the other variants compared in Table 1.
> 2) The 216M model size is based on the Pythia 169M model, which has 12 layers and 768 hidden neurons per layer, where we extended the layer to 18.
> We have added this information to section 4.1 of the paper, highlighted in blue.
>
> **Section 3.7 mentions the option to "further employ quantization techniques to convert [the activations] to integers" but it is unclear whether this was actually done in this study, and what implementation is considered in the energy consumption analysis. It should be noted that the cited references all refer to non-spiking NN works.**
>
> Response: Thank you for pointing out the confusion regarding quantization. The energy efficiency analysis is in line with the model, where activations are binarized and weights are at half-precision (FP16). Implementing quantization in SNNs is indeed feasible, and we have added citations [1,2] to reflect this. Quantized weights will reduce the computational cost from what is reported, though memory access dominates energy which will not change much between FP16 and INT16.
>
>
> [1]Shen, Guobin, et al. "Is Conventional SNN Really Efficient? A Perspective from Network Quantization." *arXiv preprint arXiv:2311.10802* (2023).
>
> [2]Venkatesh, Sreyes, Razvan Marinescu, and Jason K. Eshraghian. "SQUAT: Stateful Quantization-Aware Training in Recurrent Spiking Neural Networks." *Neuro-Inspired Computational Elements (NICE)* (2024).
>
>
> **Section 4.4 "Ablation studies" can be confusing. It compares two SNN variants to a non-spiking RWKV and an unmodified spiking RWKV. The latter is called SpikeGPT-R in this section only, making it unclear that this was the configuration used in the rest of the paper.**
>
> Response: We agree with the reviewer’s comment and have updated this section for better clarity, including an itemized list to identify the differences between model variants. "SpikeGPT-R" is now consistently referred to as SpikeGPT throughout the manuscript. The variant using a binarized FFN layer is called SpikeGPT-B, as highlighted in blue in Section 4.4.

---

> ### Author Response · Authors · 2024-05-28
> **Author Response: Part II**
>
> **Section 4.4 states "Notably, SpikeGPT-S exhibits superior training loss compared to SpikeGPT-R, indicating that spiking neurons can enhance model performance in certain cases." However, when comparing networks with LIF in SRFFN (SpikeGPT-S) instead of ReLU2 (SpikeGPT-R), training performance is at best comparable (1.109 vs. 1.113), especially without dispersion over multiple seeds reported, and test performance of SpikeGPT-S is inferior. This statement appears overblown. Also, calling this an "extensive ablation" study is inappropriate.**
>
> Response: Thank you for your careful reading. We have updated this passage to remove the mention of an extensive ablation study, referring to it instead as a "study of SpikeGPT and RWKV variants". The sentence regarding superior training loss has been removed and modified, noting that "SpikeGPT-S" is now just "SpikeGPT".
>
> The subsection heading "Ablation Study" has been changed to "A Comparison between SpikeGPT, RWKV, and variants".
>
> **The entire Section 5 "Discussion" reads as an appendix and the authors may want to consider moving it there. Section 5.1 is superficial, just noting that the membrane potential appears different in SRWKV and SRFNN layers, and that it results in SRFNN having higher firing rate. It's an interesting observation but not really discussed. Section 5.1 is also repeated verbatim in Appendix B.3. Section 5.2 consists of one paragraph discussing parallelization efforts of RWKV, which is interesting but limited in scope and not a discussion.**
>
> Response: We have moved the discussion to the appendix and integrated any discussion points and subjective views into their relevant sections for better integration with the experimental sections. Section 5.2 is now a standalone appendix section (Appendix D), and Section 5.1 has been merged into Appendix C.2 (formerly B.2, as Appendix A now describes the extensibility of LIF, as suggested by Reviewer odWR). The impact of spiking neurons on RWKV is briefly discussed in Section 4.4, and a discussion on the scaling projections of the model is delegated to a new section.
>
>
> In the end, we would like to extend our gratitude for your comprehensive and detailed review of our manuscript. Your insightful comments and suggestions have been instrumental in improving the quality and clarity of our work. In addition to the revisions mentioned in our previous response, we have made the following changes based on your specific comments:
>
> 1. The typo in Section 3.3 has been corrected. The sentence now reads, "binary spikes. This allows" instead of "binary spikes, This allows".
> 2. We have updated Figure 3 (now Figure 4) to address the confusion regarding the order of the SRFNN and SRWKV blocks. The figure now accurately reflects the architecture of SpikeGPT, with the SRWKV block on the left, followed by the SRFNN block on the right. This change ensures that the visual representation is consistent with the description in the text.
> 3. As mentioned in our previous response, Section 5.2 has been moved to the appendix (Appendix D). As a result, the issue with referencing equations in this section no longer exists, as the equations are now correctly referenced within the appendix.

---

### Author Response · Authors · 2024-05-28
**To all Reviewers**

Dear all Reviewers,

We would like to express our sincere gratitude for your thorough and insightful reviews of our manuscript. Your valuable feedback and constructive suggestions have greatly helped us improve the quality and clarity of our work. We have carefully considered each of your comments and have made substantial revisions to address your concerns.

We are pleased to inform you that we have uploaded a new version of the manuscript, which incorporates all the changes and improvements based on your feedback. To make it easier for you to identify the modifications, we have highlighted all the revised sections in **blue** throughout the document.

We have provided a point-by-point response to each of your comments, addressing the specific issues raised and explaining how we have revised the manuscript accordingly. These responses can be found in the "Author Response" sections of the review form.
Key updates include:

1. Clarification on the architectural choices for the 46M and 216M model variants, and the inclusion of layer and hidden size dimensions for the 216M model (Reviewer KfdC).
2. Improved clarity in the "Ablation studies" section, including consistent naming of model variants (Reviewer KfdC).
3. Revisions to the discussion section, moving to the appendix (Reviewer KfdC).
4. Inclusion of detailed documentation, tutorials, and open-source code to help mitigate the steep learning curve associated with training SNNs (Reviewer odWR).
5. A new subsection titled "Scaling SpikeGPT" to provide insights into the scalability challenges and experiments on larger-scale variants of SpikeGPT (Reviewer odWR).
6. An updated Appendix A with additional background material on SNNs, including the derivation of the LIF neuron and an explanation of surrogate gradient descent (Reviewer zsxD).

We would like to thank you once again for your valuable input and the time you have invested in reviewing our work. Your expertise and insights have been invaluable in improving the quality of our research. Please do not hesitate to contact us if you have any further questions or require additional clarification.

Sincerely,

Paper2554 Authors

---

### Decision · Action_Editor_HgJZ · 2024-06-07

**Recommendation:** Accept as is

**Comment:**

All reviewers unanimously recommend the acceptance of the manuscript in its present form, as (i) all claims are convincingly supported by appropriate experiments, and (ii) the overall contribution (the “first demonstration of spiking neural networks” applied to language modeling) is significant and of interest to the TMLR audience.

**Audience:**

The paper can elicit interest especially among researchers working on language models, neuromorphic computing and energy-efficient deep learning.

**Claims And Evidence:**

The paper proposes replacing the attention mechanism in the Transformer architecture of language models with a spiking neural network-based alternative, thereby improving overall complexity in terms of sequence length and energy efficiency. While some reviewers were initially skeptical about the clarity of the paper, particularly regarding the technicalities of spiking neural networks, the authors provided convincing clarifications during the discussion phase and modified the paper accordingly.
All reviewers unanimously recommend the acceptance of the manuscript in its present form, as (i) all claims are convincingly supported by appropriate experiments, and (ii) the overall contribution (the “first demonstration of spiking neural networks” applied to language modeling) is significant and of interest to the TMLR audience.